# Entrapment of Binaural Auditory Beats in Subjects with Symptoms of Insomnia

**DOI:** 10.3390/brainsci12030339

**Published:** 2022-03-02

**Authors:** Eunyoung Lee, Youngrong Bang, In-Young Yoon, Ha-Yun Choi

**Affiliations:** 1Department of Psychiatry, Veteran Health Service Medical Center, Seoul 05368, Korea; ehddbs48@naver.com; 2Department of Psychiatry, Ulsan University Hospital, Ulsan 44033, Korea; long-e@daum.net; 3Department of Neuropsychiatry, Seoul National University Bundang Hospital, Seongnam 13620, Korea; iyoon@snu.ac.kr; 4Department of Psychiatry, Seoul National University College of Medicine, Seoul 03080, Korea

**Keywords:** insomnia, binaural auditory beats, QEEG, spectral analysis

## Abstract

Binaural beat (BB) stimulation, which has two different frequencies for each ear, is reportedly effective in reducing anxiety and controlling mood. This study aimed to evaluate the brain wave entrainment effect of binaural beats and to propose an effective and safe supplementary therapy for relieving the symptoms of insomnia. Subjects between 20 and 59 years of age with subclinical symptoms of insomnia were recruited from the community. Quantitative electroencephalography was measured twice, before and two weeks after the BB intervention. Participants used the apparatus with or without 6 Hz BB for 30 min before going to bed for two weeks. When music with BB was played, the relative theta power increased (occipital, *p* = 0.009). After two weeks of intervention with music, the theta power increased when listening to music with BB (parietal, *p* = 0.009). After listening to music with BB for two weeks, the decrease in the beta power was more noticeable than after using music-only devices when participants listened to music in the laboratory (occipital, *p* = 0.035). When BB were played, the entrapment of the theta wave appeared. Therefore, exposure to music with BB is likely to reduce the hyper-arousal state and contribute to sleep induction.

## 1. Introduction

People with symptoms of insomnia account for 22.8% of the total population, and 5% of people with insomnia meet the Diagnostic and Statistical Manual of Mental Disorders, fourth edition (DSM-IV) criteria [1]. In a study based on the International Classification of Diseases, Tenth Revision (ICD-10) or the hypnotic prescription using the National Health Insurance Service—National Sample Cohort (NHIS-NSC) data, the prevalence of insomnia was found to be 5.78% [2].

Insomnia has the potential to increase the mortality rate from cardiovascular disease in relation to physiological awakening, and an objectively short duration of sleep is associated with attention shift impairment. In addition, insomnia is a risk factor for alcohol dependence and other mental disorders. Therefore, insomnia needs to be treated because appropriate treatment significantly lowers the severity of comorbidities and significantly helps to improve fatigue, anxiety, depression and the quality of life [3].

For patients with insomnia, clinic visits may be delayed as they usually take over-the-counter drugs and herbal medicines first [4,5,6]. Cognitive behavioral therapy (CBT) is the recommended initial treatment [7,8,9] and requires the patient’s time and effort. However, only limited clinics perform CBT. Drug treatment is widely available, but experts do not recommend taking benzodiazepines and z-drugs in the long term because there are no meta-analyses on the effectiveness and safety of the long-term use of sleeping pills yet [7,8,9]. In particular, when the elderly take benzodiazepine, side effects such as delirium, falls and fractures are likely to occur [10]. Although the causal relationship has not been clearly identified, studies show that the use of benzodiazepine can affect cognitive function [11,12]. Therefore, doctors should be careful when prescribing it to the elderly population.

Binaural beats (BB) are reportedly effective in enhancing arousal and memory [13], thereby reducing anxiety [14] and controlling mood conditions [15]. Therefore, the application of BB to people with symptoms of insomnia seems to be effective, though the exact mechanism has not yet been identified. Binaural auditory beats refer to a phenomenon in which sine waves of a specific frequency are formed in the brain by listening to two different frequencies in both ears. BB are central beats processed in the medial superior olivary nuclei of the pons [16,17], not the acoustic nerve of the ear. Although the exact mechanism of BB is not yet known, one of the hypotheses is the brain wave entrainment effect or frequency follow response. Entrainment effect refers to the tendency of the brain wave to change the frequency toward that of external stimuli. However, several clinical trials have reported contradictory results [18,19,20,21,22,23]. 

If people with sleep disturbances listen to the BB of the theta wave and the entrapment effect occurs, the low-frequency brain wave will increase and the high frequency brain wave will decrease. This will be effective in reducing cortical hyperarousal and inducing drowsiness, thereby reducing the symptoms of insomnia. It is already known that increased beta waves on quantified EEGs reflect cortical hyperarousal [24]. Previous studies reported that the entrainment of theta EEG with theta waves may increase hypnotic susceptibility or help sleep promotion [20,21]. In this study, we have proposed an effective and safe protocol to improve the symptoms of insomnia by observing the entrapment effect on EEGs by playing the BB of theta waves with music as the carrier.

## 2. Materials and Methods

### 2.1. Participants

From April 2017 to January 2018, 86 subjects with insomnia, aged 20 to 59 years, were recruited from the community. All the subjects had an insomnia severity index of less than 15 and only mild symptoms of insomnia that did not interfere with daily life. Twenty-one subjects who met the International Classification of Sleep Disorders—Third Edition (ICSD-3) criteria [25] for insomnia disorder, had a BDI of 20 or more or had regularly taken related drugs within the past year were excluded from this study. We further excluded seven people who withdrew their consent and two people who could not be reached. Eleven subjects additionally withdrew their consent. To exclude subjects with sleep apnea, 45 subjects underwent polysomnography. Two of them withdrew their consent. Thus, a total of 43 subjects participated in this study (Figure 1). This study was approved by the institutional review board of Seoul National University Bundang Hospital (B-1702/381-002). The baseline demographic and clinical characteristics of the participants and their outcome measures for the two weeks of this study have previously been published [26].

### 2.2. Procedure

Participants used either a test device in which music and BB of 6 Hz played simultaneously or a sham device (control group) in which only music played for 30 min before going to bed every day for two weeks. The two groups were randomly assigned according to a random assignment table. The researcher randomly selected three beat songs similar to the biorhythm and allowed the participants to choose the type of music they wanted; classical music, sounds of nature or pop songs. The subjects could not hear or distinguish the BB sound. 

An independent person (belonging to the Medical Research Cooperation Center at Seoul National University Bundang Hospital, Seongnam, Korea) who was not directly related to this clinical trial stratified gender and age into 10 blocks using a block randomization method and prepared a random assignment table so that the test and control groups could be assigned in a ratio of 1:1. In order to minimize bias, this study prevented researchers from seeing the random assignment numbers of all the subjects. Compliance was defined as listening to music for more than 10 days during the study period. Researchers recorded all the adverse reactions that occurred using the code specified in the World Health Organization Adverse Reaction Terminology (WHO-ART) version 92 updated by the Upsala Monitoring Center.

### 2.3. Quantitative Electroencephalography (QEEG)

This test proceeded non-invasively and painlessly. The EEG signals of the participants were measured using the 64-channel NeuroScan SynAmps device (Compumedics, Charlotte, NC, USA). The participants were in an electromagnetically shielded room with a comfortable sitting position and a cap attached to the electrodes on their head. The examiner continuously checked if the participant was sleepy during the examination. EEG electrodes were attached according to the international 10–20 system. Referential montages and 16 EEG electrodes were used for the analysis (FP1, FP2, F3, F4, F7, F8, C3, C4, P3, P4, T3, T4, T5, T6, O1, O2, average references). Recording of EEGs started when the electrical impedance of all electrodes was below 5 kΩ, and EEG signals were sampled at 1000 Hz and digitalized. The high pass filter was set to 100 Hz with the low pass filter set to 0.3 Hz. 

The frequency bands were defined as delta (1.0–4.0 Hz), theta (4.0–8.0 Hz), alpha (8.0–12.0 Hz) and beta (12.0–25.0 Hz). The five brain regions were defined as the frontal portions (FP1, FP2, F3, F4, F7, F8), central portions (C3, C4), head portions (P3, P4), temporal portions (T3, T4, T5, T6) and occipital portions (O1, O2).

### 2.4. Spectral Analysis

We selected 100 to 120 s of the EEG recordings without artifacts (e.g., muscle activity, small body movements, eyelid movements and micro-sleep) for each of the five steps of the protocol. Using Neuroguide^®^ software, the fast Fourier transformation was used for the spectral analysis of the QEEG data. We selected artifact-free recording by visual analysis with the artifact rejection toolbox of Neuroguide^®^ software [27]. The extracted frequency bands were delta (0.5–4 Hz), theta (4.0–8.0 Hz), alpha (8.0–12.0 Hz), low beta (12.0–14.0) and beta (14.0–25.0 Hz) oscillations. The spectral density at low frequencies reflected the degree of sleep promotion, while the spectral density in the high-frequency band measured the extent of the wakefulness promotion [28,29]. The EEG machine was set with a high-frequency filter of 35 Hz, a low-frequency filter of 0.3 Hz and a 60 Hz filter. No digital filter was applied at the time of the power calculation. The absolute spectral powers of several frequency bands were obtained, and relative power was calculated as the absolute power of each band divided by the total absolute power. 

### 2.5. Protocol

A new audio apparatus manufactured by Dlogixs (Gyeonggi-do, Korea) can induce a target brain wave by producing BB [30]. QEEG was measured twice, before and two weeks after the BB intervention period. QEEG was conducted in the lab to evaluate the brain wave entrainment effect of BB. Since the effect of EEG changes due to music could not be excluded, the effect of using the sham device, which played music without BB, was also observed. The subject was in a resting state for 5 min without any stimulation, listened to music without BB or music with BB for 5 min, and the EEG was restored to its original state (wash-out) for 5 min. The subject then listened to music with BB or only music for 5 min, maintained the EEG measure for 5 min and then finished the test. The order of music and music with BB was randomly assigned to each study subject.

The absolute amount (μV) or relative amount (%) of the alpha, beta, theta and delta wave changes in level 4 (music + BB) compared to level 1 (no sound), level 2 (music) or level 3 (resting) in an individual were analyzed by the paired t test. An independent t test analysis was performed to compare the EEG between the test group (who used a test device that played music and BB at the same time) and the control group (who used a sham device that played only music).

### 2.6. Statistical Analysis

R statistical software, version 4.0.1 (R Development Core Team; R Foundation for Statistical Computing, Vienna, Austria) was used to analyze the data. Data were presented as the mean and standard deviation. We verified normality with the Shapiro–Wilk test. A paired t test or Wilcoxon signed rank test was performed for comparing continuous variables before and after stimuli according to normality. Two sample t-tests or a Wilcoxon rank sum test was used for comparing continuous variables between two groups according to normality. We used a chi-square test to compare dichotomous variables between two groups. Bonferroni corrections were performed to correct errors that may have occurred in multiple comparisons. 

## 3. Results

### 3.1. Demographic, Clinical Characteristic and Polysomnographic Findings

A total of 43 subjects (32 females, mean age = 34.3 ± 10.4 years) participated in this study. There was no significant difference in demographics, such as age, sex and BMI, between the test and control groups. There were also no differences between the questionnaire of the Pittsburgh Sleep Quality Index (PSQI), Epworth Sleepiness Scale (ESS), Insomnia severity index (ISI), Beck depression Inventory II (BDI), State-Trait Anxiety Inventory—State (STAI-S) and the Quality of Life scale, abbreviated version (QOL-BREF), and the polysomnographic findings (Table 1).

### 3.2. Effect of Stimuli Compared to Baseline before Using Device

Before intervention, when the participants listened to music without BB in the laboratory, the relative power of the delta (temporal, *p* = 0.004; parietal, *p* = 0.005; occipital, *p* = 0.006) and theta frequencies (temporal, *p* = 0.004; central, *p* = 0.001; parietal, *p* = 0.001; occipital, *p* = 0.003) increased, while that of alpha decreased (frontal, *p* = 0.008; temporal, *p* = 0.012; central, *p* = 0.008; parietal, *p* = 0.004; occipital, *p* = 0.005). When the participants first listened to music with BB, the relative power of the theta frequency increased (occipital, *p* = 0.009) (Figure 2A and Figure 3A).

### 3.3. Effect of Stimuli Compared to Baseline after Using Device

After two weeks of intervention with music without BB, the theta power increased after listening to music with BB in the laboratory (parietal, *p* = 0.009) (Figure 2B and Figure 3B). However, the power of the EEG did not change significantly from the baseline EEG when music with or without BB was played in the laboratory. After listening to music with BB for two weeks, the decrease in the beta power from baseline was more prominent than after listening to music without BB when participants listened to music without BB in the laboratory (occipital, *p* = 0.035) (Figure 2C and Figure 3C).

## 4. Discussion

As a result of this randomized controlled trial, the relative power of the delta and theta waves of waking EEG increased and the relative power of the alpha wave decreased after listening to music for five minutes before using the device. However, only the relative power of the theta wave increased after listening to music with BB for five minutes after an intervention for two weeks. In the test group that listened to music with BB for five minutes for two weeks, the beta power decreased more significantly from the baseline than it did in the control group.

In this study, the relative power of low frequency increased and that of high frequency decreased after listening to music, suggesting that listening to music may promote relaxation and sleep. According to a study investigating the effect of music on brain waves, soothing music relaxes people with high arousal levels, and stimulating music increases the arousal level in people with low arousal levels [31]. However, previous results on the improvement of symptoms of sleep quality with music are inconsistent [32,33,34]. The results of a meta-analysis of 10 randomized studies showed an improvement in the sleep quality in patients with acute or chronic sleep disorders [35]. Studies have also used polysomnography to confirm the beneficial effects in community-dwelling adults with insomnia [34]. On the other hand, a study that used music created to promote sleep, called the ‘Delta Sleep system’, showed that the effect is insufficient to people with and without sleep disturbance [36]. Music manufacturers claim that music-induced delta EEG (0.5~3.5 Hz) improves sleep quality and increases slow wave sleep, but this has not been proved yet [36].

In the previously published paper of the findings of this study, no significant difference was seen in the insomnia severity index score between the group with the theta wave BB application and the control group, but the effect size was much stronger in the music with the BB group than in the group with music alone (Cohen d = 1.02 vs. 0.58) [26]. As previous studies have applied various stimulation durations and frequencies, it is difficult to conclude that stimulation using BB will be helpful for people suffering from anxiety disorders and insomnia by inducing changes in autonomic function. However, some studies have also derived positive results. A study showed that parasympathetic activation, sympathetic withdrawal and relaxation appeared after applying a theta frequency BB [37]. Weiland et al. reported that after applying 10 Hz of BB for 20 min, the STAI was lower than that of the control group [38]. Padmanabhan et al. reported that pre-operative anxiety evaluated by STAI after applying a delta frequency BB to patients for 30 min was lower than that of the control group [14].

In this study, the relative power of the theta wave in the parietal region selectively increased with music and BB stimulation for 5 min in subjects using the sham device. 

The alpha power steadily declines as the theta power increases throughout the transition from a resting awake state to a sleeping state. As a result of studying the resting awake EEG of sleep deprivation subjects, subjective sleepiness was negatively related to the entire alpha and positively related to central frontal theta frequencies [39]. In terms of the increase in the theta power of the parietal region when BB is applied, the possibility that this BB induces sleepiness can be seen. In this study, there was a difference in the frontal area, but it was eliminated following Bonferroni correction. When undertaking cognitive tasks that require focused attention or practicing meditation, the frontal midline theta rhythm arises [40,41,42]. In a prior study, theta activity, including the frontal midline theta rhythm, was shown to be inducible in the frontal and parietal-central regions within 10 min after an exposure with 6 Hz of BB for 30 min [43]. Differences in the duration of BB exposure or the time between exposure and measurement may have an impact on the outcome.

The baseline EEG of the test group who listened to music with BB for 2 weeks did not show a difference compared to the control group, which only listened to music, which may be due to the habituation effect by prolonged exposure. In a previous study on 27 healthy young adults, 5 received pink noise-only training, and 22 received the BB training. As a result of the BB training at 5.5 to 8 Hz for four hours, there was no difference in the brain waves compared to that of the control group. However, when pink noise was heard for 40 min, the theta power increased. These findings suggest a theta blocking effect by prolonged exposure [21]. On the other hand, when BB of 7 Hz were applied for 15 min, the delta power increased in the left temporal region [23], and when 4 Hz and 6.66 Hz BB were applied for 10 min, the auditory steady state response was induced in the temporal, frontal and parietal regions [19]. 

In the experimental group that listened to music with BB for two weeks, the beta power decreased from the baseline after listening to music for five minutes compared to that seen in the control group, suggesting that exposure to theta frequency BB can reduce the hyperarousal state. A study on 13 healthy young adults stimulated with BB at 5 Hz for five minutes did not show clear EEG frequency enhancement, but similar to our findings, the relative power of the beta band decreased. Additionally, the researchers insisted that when brain connectivity was measured using cross-mutual information (CMI) during theta frequency stimulation, a decrease in CMI at first indicates a possible approach to sleep promotion [20]. Another study, in which acoustic and binaural beats of five different frequencies (4.53 Hz theta, 8.97 Hz alpha, 17.93 Hz beta, 34.49 Hz gamma or 57.3 Hz upper gamma) were applied to 14 healthy young adults for three minutes, did not show any effect on the enhancement of the EEG spectral power at those frequencies. However, since the observed power of the ANOVA test and the sample size were small, there is a possibility of the lack of positive findings [44]. 

This study was meaningful in that a new non-invasive treatment for insomnia was delivered through a randomized controlled trial. However, it has several limitations also. Due to the small sample size, statistically meaningful results could not be derived as much as expected. Using music as a carrier, it is difficult to rule out the possibility that the carrier influenced the entrapment of BB. In previous studies that reported that entrapment was observed using a BB of theta frequency, either carriers were not used [23] or 250 Hz carriers [43], 240 Hz carriers and 480 Hz carriers [19] were used. It is also possible that subjects with subclinical insomnia may have influenced the outcome. If the stimulation had been conducted on insomnia patients, more meaningful results may have been obtained.

## 5. Conclusions

When participants listened to music with a theta BB, the entrapment of the theta wave appeared. The result that theta wave increased when exposed to BB with music shows the possibility that BB can induce sleepiness. The increase in delta and theta power, as well as the decrease in alpha power, observed after listening to music is thought to alleviate tension and provide a relaxation effect. Thus, music and BB exposure are likely to reduce the hyper-arousal state and contribute to relieving insomnia. If researchers continue further research using different BB carriers and varying the duration of BB exposure, it will be possible to develop BB that can treat insomnia.

## Figures and Tables

**Figure 1 brainsci-12-00339-f001:**
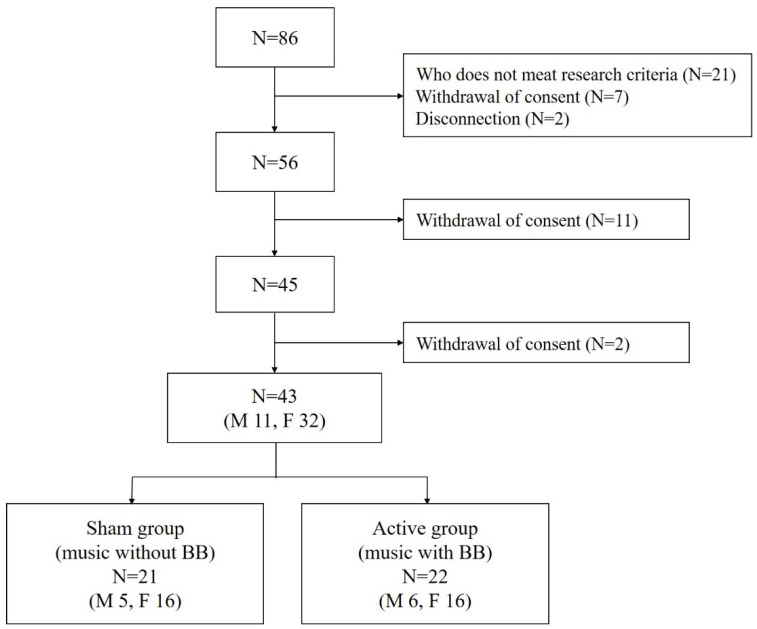
Flow chart summary of the participants. N indicates number; M, male; F, female.

**Figure 2 brainsci-12-00339-f002:**
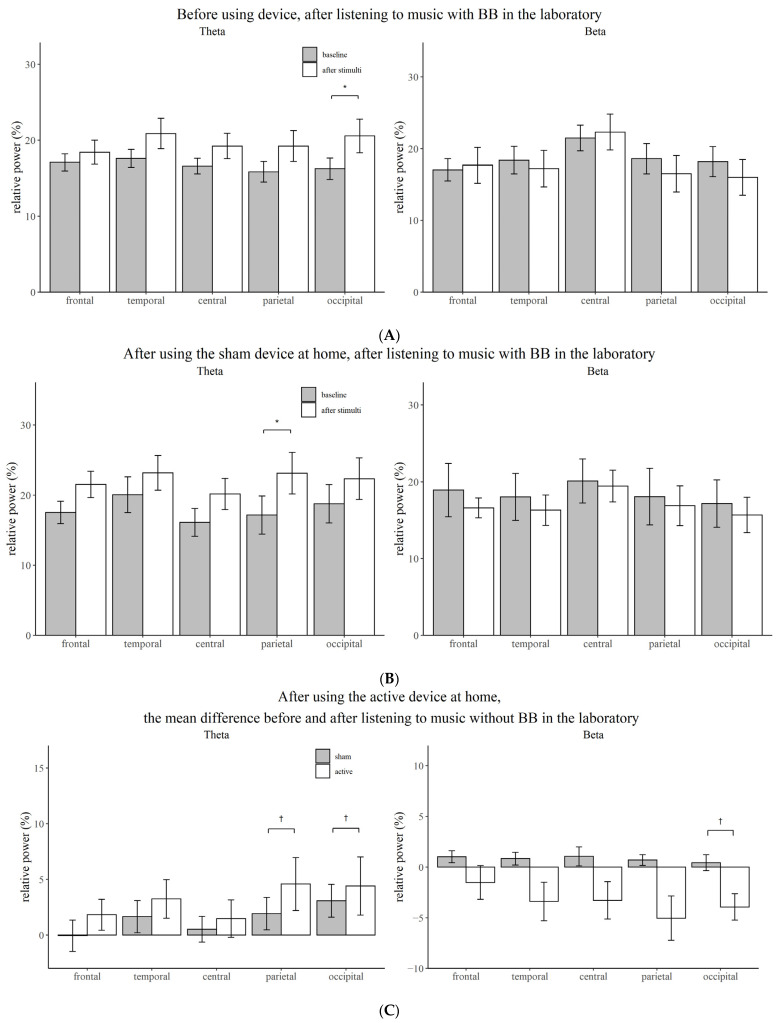
Comparison of the relative power between the baseline state and the state after stimulation for each brain region. (**A**) Before using device, after listening to music with BB in the laboratory. (**B**) After using the sham device at home, after listening to music with BB in the laboratory. (**C**) After using the active device at home, the mean difference before and after listening to music without BB in the laboratory. Vertical bars represent standard deviation; BB, binaural beat; *, Bonferroni corrected *p*-value < 0.05; †, *p*-value < 0.05.

**Figure 3 brainsci-12-00339-f003:**
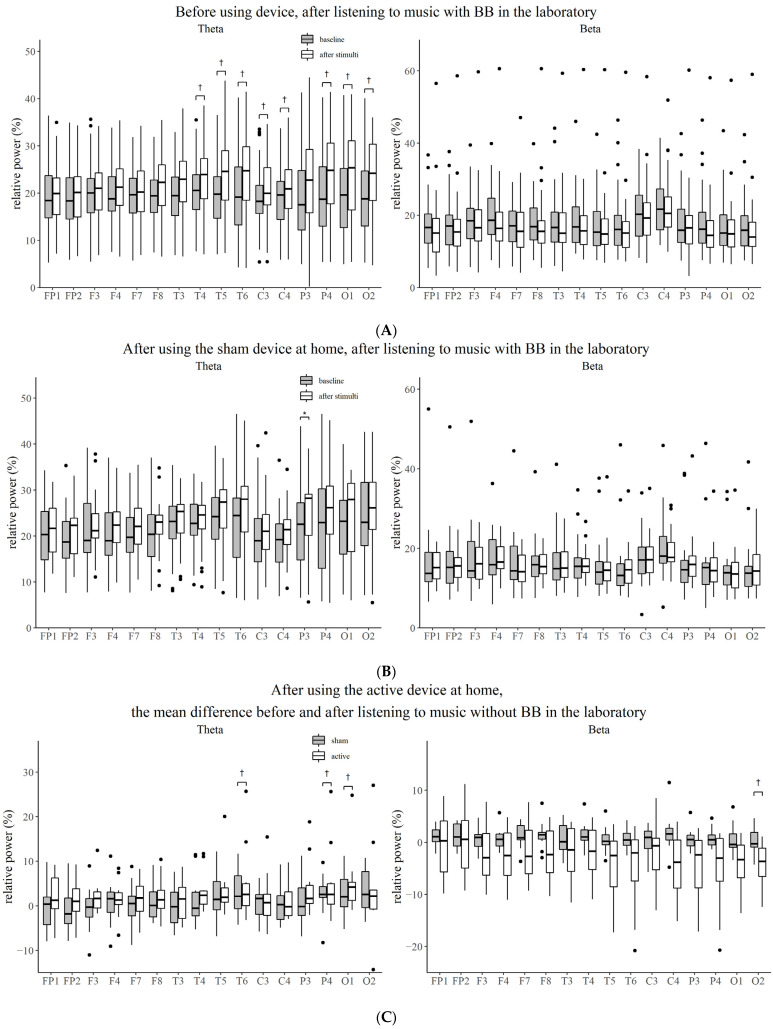
Comparison of the relative power between the baseline state and the state after stimulation for each electrode. (**A**) Before using device, after listening to music with BB in the laboratory. (**B**) After using the sham device at home, after listening to music with BB in the laboratory. (**C**) After using the active device at home, the mean difference before and after listening to music without BB in the laboratory. BB, binaural beat; *, Bonferroni corrected *p*-value < 0.05; †, *p*-value < 0.05.

**Table 1 brainsci-12-00339-t001:** Demographics and polysomnographic findings of eligible participants.

	Total (N =43)	Music (N = 23)	Music with Binaural Beat (N = 20)	*p*
Age, yr	34.3 ± 10.4	32.8 ± 9.4	35.9 ± 11.4	0.338
Sex	Men: N = 10 Women: N = 32	Men: N = 6 Women: N = 16	Men: N = 4 Women: N = 16	0.434
BMI, kg/m^2^	22.5 ± 3.0	22.0 ± 2.4	23.0 ± 3.7	0.318
PSQI score	8.7 ± 2.7	8.7 ± 2.4	8.7 ± 3.0	0.956
ESS	9.1 ± 4.1	8.9 ± 3.6	9.3 ± 4.8	0.794
ISI	9.9 ± 3.1	10.0 ± 3.1	9.7 ± 3.2	0.721
BDI	8.1 ± 4.6	8.4 ± 4.8	7.8 ± 4.5	0.632
STAI-S	39.1 ± 9.2	37.9 ± 8.8	40.5 ± 9.6	0.364
AHI, n/h	1.6 ± 3.3	0.8 ± 1.6	2.5 ± 4.5	0.121
WASO, min	57.5 ± 61.0	61.8 ± 75.2	52.6 ± 40.3	0.628
TST, min	390.7 ± 71.0	392.1 ± 78.7	389.1 ± 63.0	0.893
Sleep latency, min	23.1 ± 24.8	18.2 ± 12.2	28.7 ± 33.5	0.197
Sleep efficiency, %	82.8 ± 14.2	83.0 ± 16.2	82.6 ± 11.8	0.917
Stage 1, %	7.4 ± 4.5	7.7 ± 5.1	7.1 ± 3.8	0.674
Stage 2, %	47.2 ± 8.4	47.7 ± 8.2	46.7 ± 8.9	0.707
Stage 3, %	15.9 ± 6.2	14.8 ± 5.0	17.1 ± 7.2	0.212
REM, %	17.8 ± 5.7	18.3 ± 5.8	17.2 ± 5.8	0.537

Data are presented as mean ± SD. BMI, body mass index; PSQI, Pittsburgh sleep quality index; ESS, Epworth sleepiness scale; ISI, insomnia severity index; BDI, beck depression inventory; STAI-S, State-Trait Anxiety Inventory—State; AHI, apnea-hypopnea index; WASO, wake after sleep onset, min; TST, total sleep time; REM, rapid eye movement sleep.

## Data Availability

Not applicable.

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
