# Peer review of "Entrapment of Binaural Auditory Beats in Subjects with Symptoms of Insomnia"

_brainsci, 2022, doi:10.3390/brainsci12030339_

Round 1

Reviewer 1 Report

1. Paper is well written. No grammatical errors.

2. Line number 105 needs citation.

3.Description of artifact removal method is needed.

4. Figure No 2 should be more clear. It is not readable.

5. Justification on lines 258-262 needed. Was the study not conducted on insomniac patients?

6.  Need to highlight the factors and correlation between theta power and alpha power.

7.  Specify the need of considering theta waves for insomnia.

8. Total sleep cycle duration is not mentioned and the transition phase of each sleep wave needs to be highlighted.

9. Need to mention theta power analysis for the frontal region of the brain.

10. Results are not conclusive as the duration for assessing sleep insomnia is very less. Conclusion section needs to be strengthened with the results quantified.

Author Response

  1. Paper is well written. No grammatical errors.

We appreciate your time and effort in reviewing our manuscript. We have attempted to address your concerns.

  1. Line number 105 needs citation.

When planning a study, it is typically considered that 20% of the participants or period are likely to drop out. According to this, compliance is attained when the participants utilizes a device for more than 10 days, or 80 percent of the 14-day period.

3.Description of artifact removal method is needed..

As suggested, we have added as follows. “We selected artifact-free recording by visual analysis with the artifact rejection toolbox of Neuroguide® software”

  1. Figure No 2 should be more clear. It is not readable.

We apologize for any inconvenience this has caused. As suggested, in the revised manuscript, we have revised the figure to be more readable.

  1. Justification on lines 258-262 needed. Was the study not conducted on insomniac patients?

The study was conducted on healthy patients instead of insomniac patients. For being clear, we have revised as follows. “A previous research on 13 healthy, non-insomniacs, young adults stimulated with BB at 5 Hz for five minutes did not show clear EEG frequency enhancement

  1. Need to highlight the factors and correlation between theta power and alpha power.

As suggested, we have added as follows in section 4.

“The alpha power steadily declines as the theta power increases throughout the transition from a resting awake state to a sleeping state. As a result of studying the resting awake EEG of sleep deprivation subjects, subjective sleepiness was negatively related to entire alpha and positively related to central frontal theta frequencies.”

  1. Specify the need of considering theta waves for insomnia.

For being clear, we have added this sentence in section 1.

“Previous studies reported that entrainment of theta EEG with theta wave may increase hypnotic susceptibility or help sleep promotion.”

  1. Total sleep cycle duration is not mentioned and the transition phase of each sleep wave needs to be highlighted.

`We mentioned total sleep time and durations of each sleep stage in table 1. QEEG used in this study was intended to measure brain waves during awakening. We used polysomnography only to exclude subjects with sleep apnea. For being clear, we have revised this sentence in section 4. “As a result of this randomized controlled trial, the relative power of the delta and theta waves of waking EEG increased and the relative power of the alpha wave de-creased after listening to music for five minutes before using the device.”

  1. Need to mention theta power analysis for the frontal region of the brain.

As suggested, we have added as follows in section 4.

”In this study, there was a difference in the frontal area, but it was eliminated following Bonferroni correction. When undertaking cognitive tasks that require focused attention or practicing meditation, the frontal midline theta rhythm arises [40-42]. In a prior study, theta activity, including the frontal midline theta rhythm, was shown to be inducible in the frontal and parietal-central regions within 10 minutes after an exposure with 6 Hz of BB for 30 minutes [43J. Differences in the duration of BB exposure or the time between exposure and measurement may have an impact on the outcome.”

  1. Results are not conclusive as the duration for assessing sleep insomnia is very less. Conclusion section needs to be strengthened with the results quantified.

The participants in this study were on average 34 years old and reported fairly mild insomnia symptoms. As a result, objective indices like TST from polysomnography did not accurately reflect the individuals' insomnia. We did not reflect demographics or polysomnographic findings into the EEG results because there were no differences between participants who used active and sham devices (Table 1).

Reviewer 2 Report

In the manuscript titled “Entrapment of binaural auditory beats in subjects with symptoms of insomnia disorder,” the authors attempt to address insomnia. First of all the authors should be congratulated, they deal with a topic of great interest for public health. The manuscript is interesting but I would like to provide some feedback.

In section 2 I would suggest adding information about:

  1. Description of the EEG acquisition system
  2. Sampling rate and Impedances values, type of EEG montage, and number of channels
  3. In the paper, the .fft function is mentioned but nothing is said about the power function used. In this regard I would like to suggest the following paper: “A Machine Learning Approach Involving Functional Connectivity Features to Classify Rest-EEG Psychogenic Non-Epileptic Seizures from Healthy Controls https://doi.org/10.3390/s22010129”
  4. I would like to suggest reporting a Power function description ad information on frequencies band extraction as well features dataset organization.
  5. In section 2.5 the authors use 120 ms of clean signal and then apply the .fft (note this is not the power). I would like to suggest using the whole  EEG data, eliminating the artifacts, and then applying the spectral power (e.g on 1-sec of EEG windows (in the end 5 windows)).
  6. I would like to suggest adding information on how the test group and control group are divided?
  7. Figure 1 is in low resolution
  8. Figure 2 is not evaluable; the resolution is too low and the image is blurry. This means that the results cannot be evaluated. I would suggest calculating the power and plotting it by EEG sensor as reported in the paper “ A Machine Learning Approach Involving Functional Connectivity Features to Classify Rest-EEG Psychogenic Non-Epileptic Seizures from Healthy Controls https://doi.org/10.3390/s22010129 ”.
  9. In line 208 “Music manufacturers claim that entraining EEG” What does it mean?
  10. In line 265 “nonspecific relaxation effect.” What does it mean?
  11. In the conclusions, it is suggested to correlate the use of noise, music, the effect of relaxation as well sleep with the power. This concept must be well addressed because it is the cornerstone of the work. 

I hope that with these few comments the authors can improve a very interesting work.

Author Response

Reviewer 2:

In the manuscript titled “Entrapment of binaural auditory beats in subjects with symptoms of insomnia disorder,” the authors attempt to address insomnia. First of all the authors should be congratulated, they deal with a topic of great interest for public health. The manuscript is interesting but I would like to provide some feedback.

We are grateful for taking the time to read the manuscript and for giving us such excellent feedback. We have made every effort to improve this article by reflecting your feedback as much as possible.

In section 2 I would suggest adding information about:

  1. Description of the EEG acquisition system

As suggested, we have revised as follows.

 “The brain waves EEG signals of the participants were measured using the 64-channel NeuroScan SynAmps device (Compumedics, Charlotte, NC, USA).”

  1. Sampling rate and Impedances values, type of EEG montage, and number of channels

As suggested, we have revised as follows.

“Recording of EEG started when electrical impedance of all electrodes was below 5 kΩ and EEG signals were sampled at 1000 Hz and digitalized.”

“Referential montages and 16 EEG electrodes were used for the analysis.”

  1. In the paper, the .fft function is mentioned but nothing is said about the power function used. In this regard I would like to suggest the following paper: “A Machine Learning Approach Involving Functional Connectivity Features to Classify Rest-EEG Psychogenic Non-Epileptic Seizures from Healthy Controls https://doi.org/10.3390/s22010129”

Thank you for introducing a paper that explains the FFT function in detail. I used Neuroguide software. I moved contents about Neuroguide® software from in sections 2.5. to 2.4. , and I wrote a reference “Thatcher, R. W., Biver, C. J., & North, D. M. (2002). Real-Time Z Scores.” related to the power function.

  1. I would like to suggest reporting a Power function description ad information on frequencies band extraction as well features dataset organization.

I added a reference related to the power function of Neuroguide® software. For being clear, we have revised as follows. “The extracted frequency bands were delta (0.5-4 Hz), theta (4.0-8.0 Hz), alpha (8.0-12.0 Hz), low beta (12.0-14.0), and beta (14.0-25.0 Hz) oscillations.”

  1. In section 2.5 the authors use 120 ms of clean signal and then apply the .fft (note this is not the power). I would like to suggest using the whole EEG data, eliminating the artifacts, and then applying the spectral power (e.g on 1-sec of EEG windows (in the end 5 windows)).

As described in Section 2.5, we took EEG for 5 minutes for each state, except for 1 minute before and after, for 3 minutes, artifacts comprised muscle activity, small body movements, eyelid movements and micro-sleep were removed and used for analysis. It seems that almost whole EEG data without artifacts was used for analysis.

  1. I would like to suggest adding information on how the test group and control group are divided?

In section 2.2 we have described a block randomization method to divide the test group and control group. For being clear, we have revised as follows. “The two groups were randomly assigned according to a random assignment table”

  1. Figure 1 is in low resolution

We apologize for any inconvenience this has caused. In the revised manuscript, we have changed the figure to be more readable.

  1. Figure 2 is not evaluable; the resolution is too low and the image is blurry. This means that the results cannot be evaluated. I would suggest calculating the power and plotting it by EEG sensor as reported in the paper “ A Machine Learning Approach Involving Functional Connectivity Features to Classify Rest-EEG Psychogenic Non-Epileptic Seizures from Healthy Controls https://doi.org/10.3390/s22010129 ”.

We apologize for any inconvenience this has caused. In the revised manuscript, we have revised the figure to be more readable.

As suggested, I calculated the power and plotting it by EEG sensor as reported in the paper. (figure 3) In the process of reviewing statistics again, information related to statistical analysis was added to the text, and errors related to fig2C were corrected.

  1. In line 208 “Music manufacturers claim that entraining EEG” What does it mean?

We apologize for any inconvenience this has caused. We have revised as follows. “Music manufacturers claim that music induced delta EEG improves sleep quality~.”  

  1. In line 265 “nonspecific relaxation effect.” What does it mean?

We used the phrase 'nonspecific' since music induced several alterations in brain waves, but we removed it from the revised text because it could be confusing.

  1. In the conclusions, it is suggested to correlate the use of noise, music, the effect of relaxation as well sleep with the power. This concept must be well addressed because it is the cornerstone of the work.

As suggested, we have revised as follows.

“When participants listened to music with theta BB, the entrapment of the theta wave appeared. and the music was presumed to The result that theta wave increased when ex-posed to BB with music shows the possibility that BB can induce sleepiness. The increase in delta and theta power, as well as the decrease in alpha power, observed after listening to music is thought to alleviate tension and provide have a nonspecific relaxation effect. After exposure to music with BB for two weeks, beta power decreased more than when exposed to music without BB.  Thus, music and BB exposure are likely to reduce the hyper-arousal state and contribute to relieving insomnia. If researchers continue further research using different BB carriers and varying the duration of BB exposure, it will be possible to develop BB that can treat insomnia.”

I hope that with these few comments the authors can improve a very interesting work.

We deeply appreciate your helpful suggestions.

Round 2

Reviewer 2 Report

The authors are to be congratulated, they did a great job of revising the manuscript.